# Genetic Ablation of Pyruvate Dehydrogenase Kinase Isoform 4 Gene Enhances Recovery from Hyperoxic Lung Injury: Insights into Antioxidant and Inflammatory Mechanisms

**DOI:** 10.3390/biomedicines12040746

**Published:** 2024-03-27

**Authors:** Keisuke Watanabe, Akie Kato, Hiroyuki Adachi, Atsuko Noguchi, Hirokazu Arai, Masato Ito, Fumihiko Namba, Tsutomu Takahashi

**Affiliations:** 1Department of Pediatrics, Graduate School of Medicine, Akita University, Akita 010-8543, Japan; k.watanabe.ped@med.akita-u.ac.jp (K.W.); prdfmyfmly1003@gmail.com (A.K.); cfa69550@syd.odn.ne.jp (H.A.); atsuko@doc.med.akita-u.ac.jp (A.N.); tomy@med.akitau.ac.jp (T.T.); 2Department of Neonatology, Akita Red Cross Hospital, Akita 010-1495, Japan; arahiro@med.akita-u.ac.jp; 3Department of Pediatrics, Saitama Medical Center, Saitama Medical University, Kawagoe 350-8550, Japan; nambaf@saitama-med.ac.jp

**Keywords:** bronchopulmonary dysplasia, pyruvate dehydrogenase kinase isoform 4, hyperoxia

## Abstract

Background: Pyruvate dehydrogenase kinase isoform 4 (PDK4) plays a pivotal role in the regulation of cellular proliferation and apoptosis. The objective of this study was to examine whether the genetic depletion of the PDK4 gene attenuates hyperoxia-induced lung injury in neonatal mice. Methods: Neonatal PDK4−/− mice and wild-type (WT) mice were exposed to oxygen concentrations of 21% (normoxia) and 95% (hyperoxia) for the first 4 days of life. Pulmonary histological assessments were performed, and the mRNA levels of lung PDK4, monocyte chemoattractant protein (MCP)-1 and interleukin (IL)-6 were assessed. The levels of inflammatory cytokines in lung tissue were quantified. Results: Following convalescence from neonatal hyperoxia, PDK4−/− mice exhibited improved lung alveolarization. Notably, PDK4−/− mice displayed significantly elevated MCP-1 protein levels in pulmonary tissues following 4 days of hyperoxic exposure, whereas WT mice showed increased IL-6 protein levels under similar conditions. Furthermore, neonatal PDK4−/− mice subjected to hyperoxia demonstrated markedly higher MCP-1 mRNA expression at 4 days of age compared to WT mice, while IL-6 mRNA expression remained unaffected in PDK4−/− mice. Conclusions: Newborn PDK4−/− mice exhibited notable recovery from hyperoxia-induced lung injury, suggesting the potential protective role of PDK4 depletion in mitigating lung damage.

## 1. Introduction

Bronchopulmonary dysplasia (BPD) is a chronic pulmonary ailment that often manifests following oxygen inhalation therapy or mechanical ventilation, primarily affecting premature infants. This condition, which is associated with poor neurodevelopmental and medical outcomes, poses significant challenges in neonatal care. The pathophysiology of BPD is multifaceted and likely involves a combination of genetic predisposition, perinatal inflammation, exposure to environmental toxins and various other contributing factors. Neonates with BPD often require respiratory support, including supplementary oxygen and mechanical ventilation, to manage their condition effectively [1,2].

With significant advancements in the long-term prognosis of extremely premature infants, there has been a notable increase in the incidence of bronchopulmonary dysplasia (BPD). Given this trend, it becomes imperative to identify the risk factors linked to BPD in premature infants and explore more efficacious prophylactic and treatment strategies aimed at preventing health deterioration and alleviating serious sequelae. As the survival rates of preterm infants have improved, procuring tissue samples from fatal cases of BPD has become increasingly arduous. Hence, comprehensive observations in animal models mimicking BPD-like pathological features are indispensable for advancing our comprehension of BPD pathology.

Four distinct pyruvate dehydrogenase kinase (PDK) isozymes (PDK1, 2, 3 and 4) have been identified in humans and rodents, each exhibiting tissue-specific expression patterns. Among these, PDK isoform 4 (PDK4) plays a crucial role, as it localizes within the mitochondrial matrix of cells [3]. PDK4 functions to regulate pyruvate influx into the tricarboxylic acid cycle by inhibiting pyruvate dehydrogenase activity, thereby shifting cellular energy metabolism from mitochondrial oxidative phosphorylation to cytoplasmic glycolysis. Moreover, PDK4 modulates cellular proliferation and apoptosis by influencing NF-κB and TNF signaling pathways, which are pivotal in inflammatory cascades [4]. Notably, PDK4 demonstrates elevated expression levels in cardiac, pancreatic islet and skeletal muscle tissues [3]. The dysregulation of PDK4 expression has been implicated in various diseases due to its impact on mitochondrial function [5,6,7]. 

The overexpression of PDK4 has been linked to the exacerbation of cardiovascular disorders [8,9]. Consequently, ongoing research efforts are exploring the potential therapeutic benefits of PDK4 inhibitors in addressing these conditions [10,11]. Furthermore, basic research has uncovered the involvement of PDK4 in various aspects of acute lung injury [12,13,14].

Kimura et al. reported a 45% reduction in overall pyruvate dehydrogenase (PDH) activity in the lungs of neonatal rats exposed to 100% oxygen during the initial 6 days of life compared with normoxic controls [15]. Additionally, Tanaka et al. reported changes in specific metabolites associated with glycolysis and gluconeogenesis in the lungs of adult mice following hyperoxia exposure. They noted that this alteration was likely linked to the decreased PDH activity attributed to the upregulation of Pdk4 mRNA under hyperoxic conditions [16].

Therefore, our study aimed to investigate whether the genetic deletion of the PDK4 gene could mitigate hyperoxia-induced lung injury in newborn mice. This research endeavor seeks to shed light on potential therapeutic strategies for alleviating the burden of BPD and improving neonatal respiratory outcomes, thereby enhancing the quality of care for premature infants.

## 2. Materials and Methods

### 2.1. Animals

All procedures and protocols received ethical approval from the Animal Care and Use Committee of Akita University (Permit no. b-1-0435). PDK4−/− mice, who had undergone at least 12 generations of repeated backcrossing with C57BL/6 mice to ensure genetic stability and consistency, were housed in the animal facility at Akita University under controlled conditions. Genotyping of the mice was accomplished via PCR analysis of tail biopsies, following established protocols to confirm the absence of the PDK4 gene. These measures were taken to maintain the integrity of the experimental model and ensure the validity of the research outcomes.

### 2.2. Neonatal Hyperoxic Exposure and Recovery

This study utilized high-concentration oxygen exposure models to replicate BPD. High-concentration oxygen exposure has been a longstanding method for inducing BPD, particularly in neonatal rodents born during the saccular stage of lung development. Studies have shown that this model results in fibroblast proliferation, collagen deposition and impaired alveolarization, closely resembling the features of human BPD. Furthermore, this model offers cost-effectiveness and reproducibility with persistent pathological changes in the alveoli.

Neonatal pups were randomly allocated to either normoxia (room air) or hyperoxia (95% O_2_) groups. Hyperoxic exposure lasted for 4 days within a specialized chamber (BioSpherix, Redfield, NY, USA) equipped for continuous monitoring and the regulation of oxygen and carbon dioxide levels. To simulate physiological fluctuations, dams underwent alternating periods of normoxia and hyperoxia every 24 h. Throughout the experimental period, the chamber maintained atmospheric pressure and the mice adhered to a consistent 12 h light–dark cycle, ensuring standardized environmental conditions for the study.

### 2.3. Lung Tissue Collection

Mice were anesthetized with an intraperitoneal injection of pentobarbital (50 mg/kg) to ensure humane handling during the procedure. Following anesthesia, the pulmonary artery was cannulated and perfused with phosphate-buffered saline to remove blood from the pulmonary circulation. The right lung was then swiftly excised and immediately snap-frozen in liquid nitrogen to preserve RNA and protein integrity for subsequent analyses.

Simultaneously, the left lung was gently inflated through the trachea using 10% neutral-buffered formalin (Sigma-Aldrich, St. Louis, MO, USA) at a standardized pressure of 25 cm of gravitational force, ensuring the optimal fixation of the tissue. After one minute of fixation, the trachea was securely ligated and the lung was carefully dissected and immersed in formalin overnight at 4 °C to achieve complete tissue fixation.

Subsequently, the fixed lung tissue was dehydrated and embedded in paraffin wax to provide structural support for sectioning. Thin sections, approximately 5 mm in thickness, were then prepared using a microtome and mounted onto glass slides for histological analysis. These meticulously prepared tissue sections served as the basis for the further microscopic examination and evaluation of lung morphology and pathology.

### 2.4. Lung Histology and Morphometry

Computer-assisted morphometric analysis was conducted utilizing ImageJ software version 1.49 (NIH, Bethesda, MD, USA) to meticulously evaluate distal airspace maturation. This comprehensive evaluation focused on the following two crucial parameters: mean linear intercept (Lm) and secondary septa count. To commence the assessment, paraffin-embedded lung tissue sections were meticulously prepared and subjected to hematoxylin and eosin staining to facilitate visualization under light microscopy. Lm, a pivotal metric representing the average length of line segments spanning the airspace between alveolar surface intersections, was determined with precision [17]. The Lm measurements were meticulously performed in six distinct, non-overlapping lung parenchymal areas within a single tissue section for each animal. Each examination meticulously involved five animals per condition to ensure robust statistical analysis. Lm values were meticulously calculated by dividing the total line length across the lung section by the total intercept count, with the objective of achieving 50 intercepts per field for enhanced accuracy. Furthermore, elastin staining was conducted utilizing an elastic stain kit (Abcam, Cambridge, MA, USA) following the manufacturer’s standardized protocol. Secondary septa count, characterized by the presence of elastin, was meticulously recorded across six non-overlapping lung parenchymal areas within a single tissue section for each animal. Similar to the Lm assessment, each condition evaluated involved five animals per time point to ensure reliable and comprehensive data analysis. The evaluators conducted their assessment in a blind manner, ensuring unbiased evaluation.

### 2.5. RNA Extraction and Quantitative RT-PCR Analysis

Total RNA was meticulously extracted from five lung tissue samples per group utilizing a standardized protocol [18]. Following extraction, 500 ng of RNA underwent reverse transcription utilizing the High-Capacity cDNA Reverse Transcription Kit from Applied Biosystems to generate cDNA. The resulting cDNA was subsequently utilized in PCR reactions, employing primers specific for PDK4, interleukin (IL)-6 and monocyte chemoattractant protein (MCP)-1 sourced from Applied Biosystems, in conjunction with TaqMan Universal PCR Master Mix. The LightCycler^®^ 480 system (Roche, Basel, Switzerland) was meticulously employed for analyses, with relative mRNA expression levels determined utilizing the comparative critical threshold method. These expression levels were accurately normalized to beta-glucuronidase (Applied Biosystems, Thermo Fisher Scientific, Waltham, MA, USA, Tokyo, Japan) for robust and reliable quantification. It is acknowledged that the expression of commonly used endogenous controls fluctuated in response to oxygen. Therefore, in this study, we utilized beta-glucuronidase as the housekeeping gene, owing to its minimal susceptibility to oxygen-induced alterations.

### 2.6. Measurement of Cytokine Levels in Lung Tissue

The levels of various cytokines, including IL-6, IL-10, IL-12, tumor necrosis factor (TNF)-α, interferon (IFN)-γ and MCP-1 in lung tissue homogenates were meticulously assessed using the BD cytometric bead array mouse inflammation kit sourced from BD Biosciences, headquartered in San Diego, CA, USA. The selection of these cytokines for this study was based on their reported association with hyperoxic lung injury [19,20,21]. To perform the analysis, samples were meticulously prepared and analyzed using a state-of-the-art BD FACSAriaIII flow cytometer, also from BD Biosciences, ensuring precise and accurate measurement of cytokine levels in the lung tissue homogenates.

### 2.7. Statistical Analysis

Statistical analyses were meticulously conducted utilizing SPSS software version 28.0 (SPSS, Chicago, IL, USA). Continuous variables were meticulously expressed as the mean ± SEM or as box-and-whisker plots indicating the median, interquartile range and range. Group comparisons were conducted employing the following robust statistical methods: one-way ANOVA followed by Tukey’s post hoc test for parametric data or the Kruskal–Wallis test followed by Mann–Whitney’s U-test for nonparametric data, ensuring the reliability and accuracy of the findings.

## 3. Results

### 3.1. At 4 Days of Age, Neonatal Hyperoxia Compromised Alveolar Development in Both Wild-Type (WT) and PDK4−/− Mice, Suggesting a Deleterious Impact on Lung Maturation Regardless of PDK4 Status

Under physiological room air conditions, mice exhibited meticulously structured terminal airways, indicative of typical pulmonary development. However, when neonatal mice were subjected to hyperoxia for 96 h, a hindrance in alveolar development ensued, characterized by the expansion and simplification of alveoli (Figure 1a). Furthermore, hyperoxic exposure suppressed secondary septation, exacerbating the impediment to proper lung maturation (Figure 1b). Remarkably, both WT and PDK4−/− mice exposed to neonatal hyperoxia manifested a significant augmentation in Lm compared to normoxic controls (Figure 1c), signifying enlarged alveolar spaces. Similarly, the count of secondary septa significantly diminished following neonatal hyperoxia in both WT and PDK4−/− mice (Figure 1d), indicative of impaired alveolar septation. Nonetheless, no notable disparities were discerned in the Lm, or the secondary septa count between WT and PDK4−/− mice at 4 days of age when exposed to either normoxia or hyperoxia, suggesting comparable responses to oxygen exposure in terms of lung morphology.

### 3.2. The Compromised Alveolar Development Noted in PDK4−/− Mice at 4 Days of Age Exhibited Substantial Recovery by the Age of 14 Days, Suggesting the Presence of a Potential Compensatory Mechanism or Inherent Resilience in Lung Maturation over Time

Following a 4-day hyperoxic exposure during the neonatal phase, WT mice exhibited a persistent elongation of Lm and a reduction in secondary septa, even after a subsequent 10-day recovery period in room air compared to normoxic counterparts at 14 days of age. In contrast, despite experiencing impaired alveolarization following neonatal hyperoxia at 4 days of age, PDK4−/− mice demonstrated a notable improvement in Lm and increased secondary septa count following room air recovery at 14 days of age, relative to similarly exposed WT mice (Figure 2). This underscores that the absence of PDK4 confers enhanced resilience to hyperoxia-induced lung injury and facilitates more robust recovery during the convalescent phase.

### 3.3. Neonatal Hyperoxia Induced Elevated Lung mRNA Expression of PDK4 in WT Mice Compared to Those Exposed to Normoxia, Indicating the Potential Role of Hyperoxia in Upregulating PDK4 Expression in the Lung Tissue of WT Mice

The mRNA expression of PDK4 in lung tissues of WT mice exhibited a significant increase following exposure to hyperoxia compared to WT mice exposed to normoxia. However, the between-group differences in this regard were not statistically significant on day 14 (Figure 3). The transient upregulation of PDK4 expression in response to hyperoxia underscores its involvement in the adaptive response to oxidative stress in the lungs. Furthermore, the normalization of PDK4 expression levels by day 14 suggests the resolution of the acute response to hyperoxia and the restoration of homeostasis in the lung tissue.

### 3.4. The Lungs Exposed to Hyperoxia Exhibited Significant Disparities in IL-6 and MCP-1 Protein and mRNA Expression Levels Compared to Those Exposed to Normoxia

To thoroughly assess the inflammatory response during hyperoxic exposure, we meticulously quantified inflammatory cytokine levels in the lungs of 4-day-old WT and PDK4−/− mice exposed to either normoxia or hyperoxia. Our comprehensive analysis unveiled no significant disparities in the protein levels of TNF-α, IFN-γ, IL-10, or IL-12 between WT and PDK4−/− mice post-normoxic or hyperoxic exposure. However, a striking observation emerged as MCP-1 levels in the pulmonary tissues of PDK4−/− mice exhibited a marked surge relative to WT counterparts following the 4-day hyperoxic exposure. In contrast, while WT mice displayed heightened IL-6 levels post-hyperoxia, PDK4−/− mice showed no discernible response to oxygen exposure (Figure 4). Subsequently, the levels of inflammatory cytokines were reassessed at 14 days of age. Interestingly, the significant differences in the expression of MCP-1 and IL-6, previously noted at 4 days of age, were no longer evident at 14 days of age. Similarly, no significant variances were noted in the other four cytokines (TNF-α, IFN-γ, IL-10, IL-12) between the two groups (Figure 5). These intriguing observations suggest that the initial discrepancies observed in cytokine expression subsequent to neonatal hyperoxic exposure may attenuate over time, hinting at the potential normalization of the inflammatory response during the post-natal developmental and recuperative phase.

Given the significantly higher levels of MCP-1 in the lungs of PDK4−/− mice compared to those in WT mice, we meticulously assessed mRNA expression levels of MCP-1 via qRT-PCR. Our findings revealed that MCP-1 mRNA expression in the lungs of both WT and PDK4−/− mice substantially surged following a 4-day hyperoxic exposure relative to normoxic conditions. Furthermore, PDK4−/− mice post-neonatal hyperoxia exhibited a markedly augmented mRNA expression of MCP-1 at 4 days of age compared to WT mice. However, subsequent 10-day recovery in room air effectively restored MCP-1 mRNA expression to baseline levels in both WT and PDK4−/− mice. Notably, IL-6 mRNA expression at 4 days of age exhibited pronounced escalation in the WT group post-oxygen exposure while showing no discernible change in PDK4−/− mice. Following the 10-day room air recovery period, IL-6 mRNA expression reverted to baseline levels in both WT and PDK4−/− mice (Figure 6). These meticulous assessments shed light on the intricate dynamics of cytokine regulation in response to hyperoxic insult and subsequent recovery, emphasizing the potential role of PDK4 in modulating inflammatory pathways during lung injury and repair.

## 4. Discussion

This study aimed to comprehensively evaluate the impact of hyperoxia-induced pulmonary injury in PDK4−/− mice compared to WT mice, revealing a noteworthy alleviation of lung damage in PDK4−/− mice following a recovery period subsequent to neonatal hyperoxic exposure lasting 4 days. In particular, the restoration of typical alveolar developmental levels in PDK4−/− mice at the age of 14 days underscored the effectiveness of the convalescent phase in mitigating the adverse effects of hyperoxia. Furthermore, the analysis of IL-6 and MCP-1 protein levels in the lung tissues yielded valuable insights, suggesting that the observed recovery in PDK4−/− mice could be attributed to their heightened antioxidant and anti-inflammatory capacities relative to WT mice. This insight sheds light on the intricate mechanisms governing the protective role of PDK4 deletion in the context of neonatal hyperoxic lung injury, presenting promising avenues for therapeutic intervention and further research exploration.

Previous studies have demonstrated the increased expression of PDK4 in the heart, pancreatic islets and skeletal muscles; however, its effect on lung disorders is not clear. However, some studies have suggested a close association of PDK4 with idiopathic pulmonary fibrosis [22] and idiopathic pulmonary arterial hypertension [23]. Several factors are implicated in the causation of BPD, including exposure to hyperoxia [24,25]. Our data suggest that hyperoxia might have both a direct and an indirect effect on lung tissue due to the convergence of multiple influences.

The interplay between lung injury and IL-6 under high oxygen exposure remains contentious. While studies have suggested the protective effect of IL-6 overexpression on adult mouse lungs [26,27], contrary findings have been observed in neonatal mice. In neonatal mice subjected to hyperoxia, the pharmacological inhibition of global IL-6 signaling and IL-6 trans-signaling was shown to improve alveolarization and survival [28]. Conversely, IL-6 overexpression was shown to protect against oxidant-mediated injury when mice were exposed to hyperoxia from day three to six of life [29,30]. In the present study, PDK4−/− mice showed suppressed IL-6 production and faster convalescence from lung injury. In a previous study, treatment with opioid analgesic remifentanil reduced the levels of IL-6 in the lungs and mitigated the protein and mRNA expression of PDK4 in myocardial tissue in lipopolysaccharide-induced septic rats [31]. We posit that IL-6 and PDK4 expression exert a multifaceted influence on both lung development and injury, necessitating further exploration.

MCP-1 is a chemotactic factor for leukocytes. Okuma et al. found that MCP-1/CCR2 signaling safeguards against hyperoxia-induced lung injury by curbing inducible nitric oxide synthase induction and subsequent reactive oxygen species generation by activating alveolar macrophages [32]. However, some studies have reported contradictory findings regarding MCP-1 overexpression and its potential deleterious effects on the lungs [33,34]. In the present study, the overexpression of MCP-1 was observed even in WT mice exposed to high oxygen concentrations, with more pronounced overexpression noted in PDK4−/− mice. We hypothesized that the extent of MCP-1 expression may influence the recovery from hyperoxic injury.

Given the frequent upregulation of PDK4 in various cancers, ongoing drug research involves investigations involving Cryptotanshinone as a PDK4 inhibitor [35,36]. Studies have shown that Cryptotanshinone not only impedes the progression of pulmonary fibrosis [37] but may also have a potential therapeutic effect in interstitial lung disease [38]. Cryptotanshinone is derived from the traditional Chinese herb Salvia miltiorrhiza Bunge. Salvia miltiorrhiza Bunge has a longstanding history of medicinal use, primarily in China, and has been the subject of numerous studies [39]. While no studies have investigated the safety and efficacy of Cryptotanshinone in neonatal diseases, its potential use in conditions such as BPD should be investigated.

PDK4 demonstrates heightened expression in both cardiac and skeletal muscle tissues. Considering the well-documented associations between cardiovascular irregularities and ailments like pulmonary hypertension, which are frequently associated with BPD [40,41], one could reasonably speculate that heightened oxygen exposure may influence blood vessel development in both pulmonary and cardiac domains. Nevertheless, it is crucial to acknowledge that the lack of histopathological scrutiny of the vascular system in our present study presents a constraint, urging further inquiry to comprehensively elucidate the underlying mechanisms.

In conclusion, our study sheds light on a significant recuperation from hyperoxic lung injury during the convalescent phase in neonatal PDK4−/− mice, surpassing the recovery observed in WT mice. This rehabilitative effect appears to stem from the heightened antioxidant and anti-inflammatory activity observed in PDK4−/− mice. Furthermore, the transient modulation of proinflammatory cytokines, particularly IL-6 and MCP-1, plays a pivotal role in driving this protective response in PDK4−/− mice. These findings underscore the multifaceted mechanisms through which PDK4 deletion confers resilience against hyperoxic lung injury, offering promising insights for the development of targeted therapeutic strategies aimed at alleviating neonatal respiratory distress and enhancing lung function recovery.

## Figures and Tables

**Figure 1 biomedicines-12-00746-f001:**
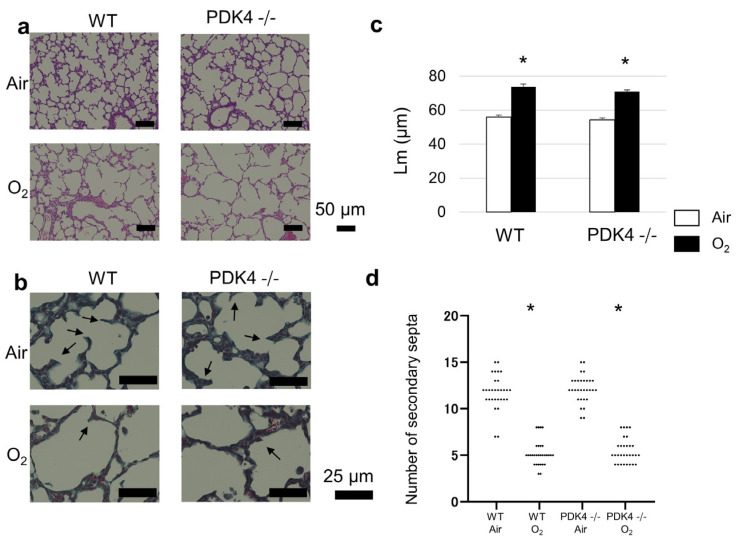
Alveolar development following neonatal hyperoxic exposure for the first 4 days of life. (**a**) Representative hematoxylin and eosin-stained histological sections from mice on day 4, illustrating alveolar morphology. Scale bar: 50 µm. (**b**) Representative elastin-stained histological sections from mice on day 4, highlighting secondary septa. Scale bar: 25 µm. (**c**) Mean linear intercept (Lm) on day 4 (n = 5 animals in each group), indicating alveolar size and complexity. Data are presented as the mean ± SEM. (**d**) Number of secondary septa on day 4 (n = 5 animals in each group), representing alveolar septation. Between-group differences were assessed by one-way ANOVA followed by Tukey’s test. * *p* < 0.05 vs. Air.

**Figure 2 biomedicines-12-00746-f002:**
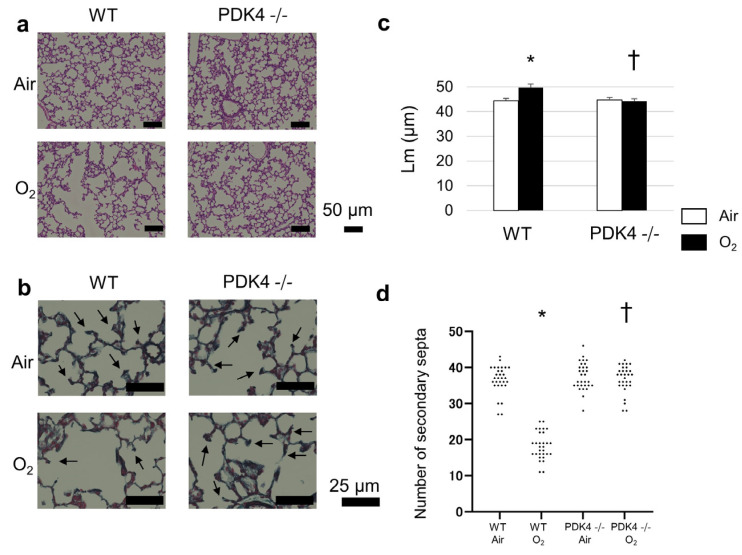
Alveolar development during recovery from neonatal hyperoxic exposure at the age of 14 days. (**a**) Representative hematoxylin and eosin-stained histological sections from mice on day 14 illustrating alveolar morphology. Scale bar: 50 µm. (**b**) Representative elastin-stained histological sections from mice on day 14, highlighting secondary septa. Scale bar: 25 µm. (**c**) Mean linear intercept (Lm) on day 14 (n = 5 animals in each group), indicating alveolar size and complexity. Data presented as mean ± SEM. (**d**) Number of secondary septa on day 14 (n = 5 animals in each group), representing alveolar septation. Between-group differences were assessed by one-way ANOVA followed by Tukey’s test. * *p* < 0.05 vs. Air, † *p* < 0.05 vs. WT O_2_.

**Figure 3 biomedicines-12-00746-f003:**
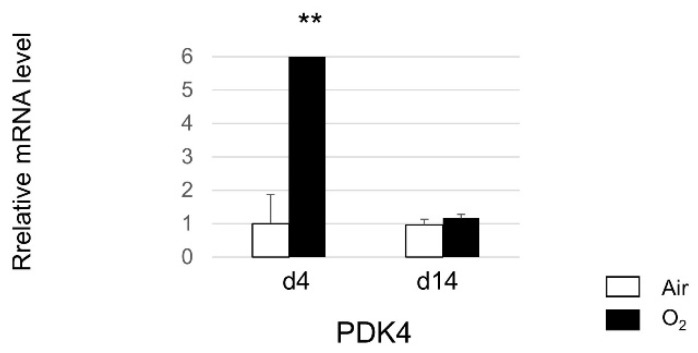
Gene expression levels of pyruvate dehydrogenase kinase 4 (PDK4) in the lungs after neonatal hyperoxic exposure. mRNA expression levels of PDK4 in lung tissues of WT mice exposed to neonatal hyperoxia are significantly higher than those in normoxia WT mice. Quantitative RT-PCR was performed on day 4 and day 14 (n = 5 in each group). Data presented as mean ± SEM. Between-group differences were assessed by one-way ANOVA followed by Tukey’s test. ** *p* < 0.01 vs. Air.

**Figure 4 biomedicines-12-00746-f004:**
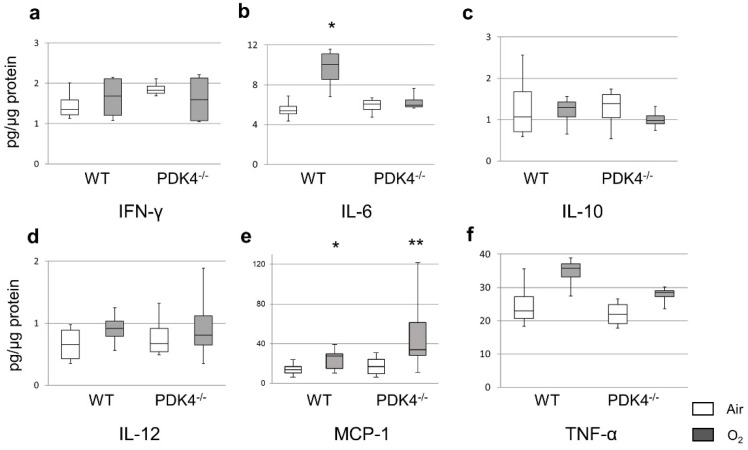
Protein levels of inflammatory cytokines in the lungs after neonatal hyperoxic exposure. The cytometric bead array was performed on day 4 (n = 5 in each group). Median protein levels of IFN-γ (**a**), IL-6 (**b**), IL-10 (**c**), IL-12 (**d**), MCP-1 (**e**) and TNF-α (**f**) in the lung homogenates. Box = 25th and 75th percentiles; bars = min and max values. Between-group differences were assessed by the Kruskal–Wallis test followed by the Mann–Whitney U-test. * *p* < 0.05 vs. Air. ** *p* < 0.01 vs. Air. IFN, interferon; IL, interleukin; MCP-1, monocyte chemoattractant protein-1; and TNF, tumor necrosis factor.

**Figure 5 biomedicines-12-00746-f005:**
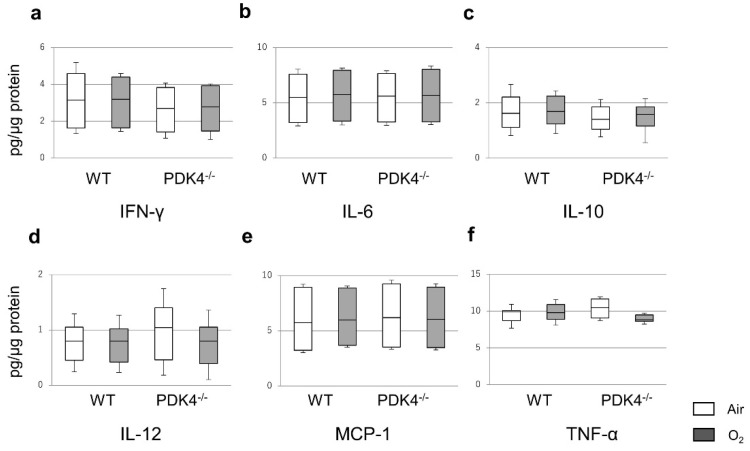
Protein levels of inflammatory cytokines in the lungs during recovery from neonatal hyperoxic exposure. Cytometric bead array was performed on day 14 (n = 5 in each group). The median protein levels of IFN-γ (**a**), IL-6 (**b**), IL-10 (**c**), IL-12 (**d**), MCP-1 (**e**) and TNF-α (**f**) in the lung homogenates. Box = 25th and 75th percentiles; bars = min and max values. Between-group differences were assessed by the Kruskal–Wallis test followed by the Mann–Whitney U-test. IFN, interferon; IL, interleukin; MCP-1, monocyte chemoattractant protein-1; and TNF, tumor necrosis factor.

**Figure 6 biomedicines-12-00746-f006:**
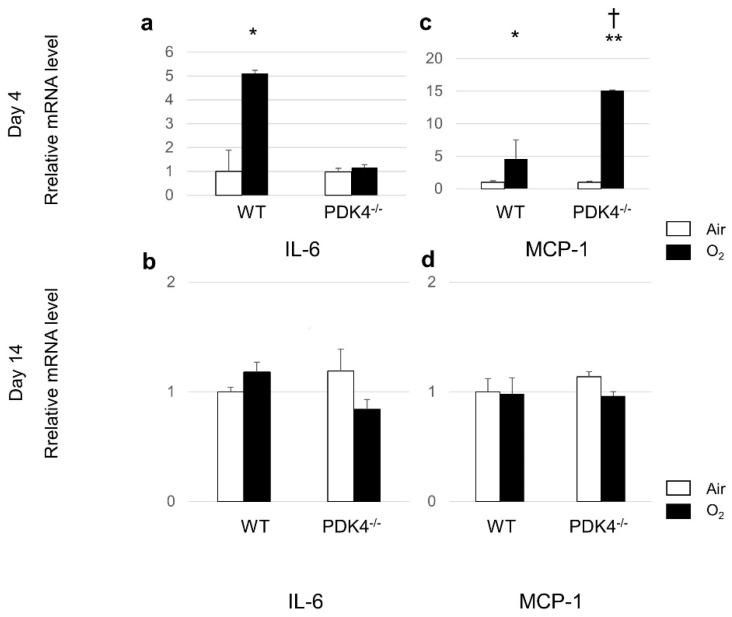
Gene expression levels of IL-6 and MCP-1 in the lungs after neonatal hyperoxic exposure. Lung mRNA expression levels of IL-6 (**a**,**b**) and MCP-1 (**c**,**d**). Quantitative RT-PCR was performed on day 4 (**a**,**c**; n = 5 in each group) and day 14 (**b**,**d**; n = 5 in each group). Data are presented as the mean ± SEM. Between-group differences were assessed by one-way ANOVA followed by Tukey’s test. * *p* < 0.05 vs. Air, ** *p* < 0.01 vs. Air, † *p* < 0.05 vs. WT O_2_. IL-6, interleukin-6; and MCP-1, monocyte chemoattractant protein-1.

## Data Availability

Data are contained within the article.

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
