# Peer review of "Genetic Ablation of Pyruvate Dehydrogenase Kinase Isoform 4 Gene Enhances Recovery from Hyperoxic Lung Injury: Insights into Antioxidant and Inflammatory Mechanisms"

_biomedicines, 2024, doi:10.3390/biomedicines12040746_

Round 1

Reviewer 1 Report

Comments and Suggestions for Authors

1. very descriptive study, I did not find a clear and extensive elaboration of the physiological processes that could be involved. for instance how and why were the specific factors (IFN-gamma, TNF alpha, IL6, IL10, IL12 and MCP1) chosen? 

2. I do not concur with the statement that an absence of PDK-4 'enhances recovery' from hyperoxic injury as basically only the number of secondary septa were significantly different, while other outcome parameters were unchanged. it may be better to present the graphs as dot-blots rather than columns with error bars as the former will give the reader a better impression of the distribution of the parameters. the zoom of the histological slides is too small, one can hardly see any detail on these images.

3. have the histological read-outs been done with a blinding of the assessor for the cohort allocation (or did they know the exact nature of the sample, e.g. D4 normoxia...)? 

4. what was the reason to only use a single housekeeping gene for the PCR analysis?

Author Response

Dear Editors and Reviewers:

We are sincerely grateful for your letter and for the thorough review of our manuscript titled " Genetic ablation of pyruvate dehydrogenase kinase isoform 4 gene enhances recovery from hyperoxic lung injury: insights into antioxidant and inflammatory mechanisms" (Ref: Submission ID biomedicines-2906443). The insightful comments provided by the reviewers have been instrumental in refining our paper and guiding our research efforts. We have meticulously reviewed each comment and made the necessary corrections and revisions, which we believe will enhance the quality and clarity of our manuscript. The main corrections in the paper and the responds to the reviewer’s comments are flowing:

Answer to reviewer 1

We really appreciate the rigorous efforts and expertise of the reviewer in reviewing our manuscript.

1.very descriptive study, I did not find a clear and extensive elaboration of the physiological processes that could be involved. for instance how and why were the specific factors (IFN-gamma, TNF alpha, IL6, IL10, IL12 and MCP1) chosen?

Response: Thank you for the feedback. It is presently understood that manifold particular elements exert an influence on BPD. Consequently, following contemplation of an assay feasible to be executed at our establishment and serving as a screening mechanism for BPD factors, we resolved to administer the examination utilizing the BD Cytometric Bead Array Mouse Inflammation Kit. Another consideration in favor of this kit's selection was the association of IFN-gamma, TNF alpha, IL-6, IL10, IL-12, and MCP-1—substances that can be assessed using this kit—with BPD, as documented in various reports. We have added the description along with relevant references in the "Materials and Methods" section.

2.I do not concur with the statement that an absence of PDK-4 'enhances recovery' from hyperoxic injury as basically only the number of secondary septa were significantly different, while other outcome parameters were unchanged. it may be better to present the graphs as dot-blots rather than columns with error bars as the former will give the reader a better impression of the distribution of the parameters. the zoom of the histological slides is too small, one can hardly see any detail on these images.

Response: Thank you for the feedback. Not only were alterations observed in the secondary septa, but also enhancements in alveolar diameter (assessed via Lm in this study) were noted in oxygen-exposed PDK4 KO mice in comparison to oxygen-exposed WT mice at the age of 14 days, as well as in the control cohort. Given the similarity of these findings to those of PDK-4, we hypothesized that the absence of PDK-4 augments recovery from hyperoxic injury. We made some adjustments to improve the zoom of the images for better visibility. Additionally, we changed the representation of the Number of secondary septa to dot-blots.

3.have the histological read-outs been done with a blinding of the assessor for the cohort allocation (or did they know the exact nature of the sample, e.g. D4 normoxia...)?

Response: Thank you for the feedback. The evaluators conducted their assessment in a blinded manner. This inclusion has been incorporated into the text accordingly.

4.what was the reason to only use a single housekeeping gene for the PCR analysis?

Response: Thank you for the feedback. It is recognized that the expression of other frequently utilized endogenous controls, such as GAPDH, fluctuates in response to oxygen, a phenomenon corroborated in our preliminary investigations. Consequently, in this study, we employed beta-glucuronidase as the housekeeping gene, given its minimal susceptibility to oxygen-induced alterations.

We tried our best to improve the manuscript and made some changes in the manuscript. These changes will not influence the content and framework of the paper. We appreciate for Editors/Reviewers’ warm work earnestly, and hope that the correction will meet with approval. Once again, thank you very much for your comments and suggestions.

Reviewer 2 Report

Comments and Suggestions for Authors

The article entitled “Genetic ablation of pyruvate dehydrogenase kinase isoform 4 gene enhances recovery from hyperoxic lung injury: insights into antioxidant and inflammatory mechanisms” has been reviewed. The authors demonstrated some promising data that PDK4 inhibition could enhance recovery from hyperoxic lung injury. However, there are some questions in this article required further explanation.

1.     From Fig. 1 and 2, it appears that the impaired alveolar development in PDK4-/- mice recovered by the age of 14 days. In Fig. 3, the mRNA expression of PDK4 in lung tissue of WT mice on Day 14 is decreased compared with Day 4. How to explain that there is no effect of PDK4 decrease on alveolar development?

2.     Fig. 4 shows the protein production of Day 4. In the previous figures about alveolar development, it recovered on day 14. Are there any data about protein production on day 14?

3.     Have you tried to administrate PDK4 inhibitor on WT mice to see the effect of alveolar recovery and protein production?

4.     I didn’t see any data about antioxidant production in this article.

Author Response

Dear Editors and Reviewers:

We are sincerely grateful for your letter and for the thorough review of our manuscript titled " Genetic ablation of pyruvate dehydrogenase kinase isoform 4 gene enhances recovery from hyperoxic lung injury: insights into antioxidant and inflammatory mechanisms" (Ref: Submission ID biomedicines-2906443). The insightful comments provided by the reviewers have been instrumental in refining our paper and guiding our research efforts. We have meticulously reviewed each comment and made the necessary corrections and revisions, which we believe will enhance the quality and clarity of our manuscript. The main corrections in the paper and the responds to the reviewer’s comments are flowing:

  1. From Fig. 1 and 2, it appears that the impaired alveolar development in PDK4-/- mice recovered by the age of 14 days. In Fig. 3, the mRNA expression of PDK4 in lung tissue of WT mice on Day 14 is decreased compared with Day 4. How to explain that there is no effect of PDK4 decrease on alveolar development?

Response: Thank you for the feedback. In our manuscript, we have demonstrated that PDK4 undergoes transient overexpression in wild-type mice in response to stress induced by exposure to elevated oxygen levels. We hypothesize that this phenomenon exerts a detrimental effect on alveolar development. Moreover, we believe that the normalization of PDK4 levels to those of the control group by day 14 signifies a crucial step in the recovery process.

2.Fig. 4 shows the protein production of Day 4. In the previous figures about alveolar development, it recovered on day 14. Are there any data about protein production on day 14?

Response: Thank you for the feedback. We have included an additional figure pertaining to the data from day 14.

3 Have you tried to administrate PDK4 inhibitor on WT mice to see the effect of alveolar recovery and protein production?

Response: Thank you for the feedback. We endeavored to administer Cryptotanshinone to neonatal mice via intratracheal or intraperitoneal routes; however, due to experimental technique limitations, we are presently unable to provide comprehensive data for submission. Moving forward, we aim to conduct further verification studies.

4 I didn’t see any data about antioxidant production in this article.

Response: Thank you for the feedback. In this study, we were unable to identify antioxidant production. Similarly to 3, this will be pursued as a future endeavor.

We tried our best to improve the manuscript and made some changes in the manuscript. These changes will not influence the content and framework of the paper. We appreciate for Editors/Reviewers’ warm work earnestly, and hope that the correction will meet with approval. Once again, thank you very much for your comments and suggestions.